# The Impact of Role Models and Mentors on the Mental and Physical Wellbeing of Sexual and Gender Minorities

**DOI:** 10.3390/bs14050417

**Published:** 2024-05-15

**Authors:** Jason Cottle, Anna L. Drozdik, Katharine A. Rimes

**Affiliations:** 1The Royal Melbourne Hospital, Melbourne, VIC 3052, Australia; 2Department of Psychology, King’s College London, London SE5 8AB, UK; anna.drozdik@kcl.ac.uk; 3Department of Psychology, Institute of Psychiatry, Psychology and Neuroscience, King’s College London, London SE5 8AB, UK; katharine.rimes@kcl.ac.uk

**Keywords:** role models, mentors, sexual and gender minorities, LGBTQ+, mental health

## Abstract

Sexual and gender minorities (SGMs) experience a higher mental health burden compared to their cisgender, heterosexual counterparts. Role models and mentors are important for wellbeing and development; however, little evidence exists exploring their impact on SGM people. This systematic scoping review identifies their association with mental and physical wellbeing. Eight databases (Medline, Embase, Cochrane CENTRAL, ERIC, Science Citation Index, Scopus, EPub and PsychInfo) were searched for eligible publications from 2000 to 2022. Two researchers identified studies, extracted data, completed quality appraisals using CASP checklists, and grouped data into outcomes relating to role model impact. From 501 citations, 12 studies *(n* = 1468 SGM people aged 15–63 years) were included. Positive role models and mentors encouraged identity acceptance through destigmatisation and positive affirmation, increased SGMs’ psychological wellbeing through improved psychological safety and self-confidence and improved their sexual health knowledge. Potential role models and mentors displaying negative behaviours could cause stigmatisation, as well as reduce identity acceptance and psychological safety. Information regarding the perceived influence of role models and mentors on substance abuse and other physical health outcomes was limited. SGMs report greater benefits from relationships with others of shared minority status, providing incentives to match mentees with role models and mentors who share or empathise with their experiences of marginalization.

## 1. Introduction

Sexual and gender minorities (SGMs) often experience higher rates of substance use and mental illness, including anxiety and depression, than their cisgender and heterosexual counterparts [1]. In the UK, lesbian, gay, bisexual, transgender, queer and other sexual and gender minority (LGBTQ+) people are almost twice as likely to experience suicidal thoughts as the cisgender and heterosexual population [2]. In the US, 47.7% of lesbian, gay or bisexual young people have considered attempting suicide compared with 13.3% of their heterosexual counterparts [3].

The vast majority of current sexual and gender minority research has investigated risk factors contributing to this health inequality and has related it to experiences of prejudice, discrimination and victimisation, with sexual and gender minorities having disproportionately higher risks of such experiences [4,5]. Meyer’s minority stress model sought to explain that the stressful effect of such experiences, in addition to other factors such as internalised stigma, may culminate in unique disparities in the mental and physical health of sexual minority people [6,7]. Notably, those with intersectional marginalised identities (including LGBTQ+ and ethnicity, race or disability) are likely to suffer worse outcomes [8]. By identifying these harmful risk factors, this research has major implications for designing practical interventions to reduce such experiences and help protect against and help address their harmful effects. Therefore, it is important to investigate positive and protective factors for mental and physical wellbeing in this group.

Role models are broadly defined as motivational people who may inspire individuals to achieve their goals and are generally regarded as important for the wellbeing and development of young people, as they provide positive representation, encourage the acquisition of new skills and stimulate goal adoption [9]. Although typically seen as positive figures, role models may also exhibit negative or socially unacceptable behaviours or attitudes. Although the majority of current research investigates the impact of role models and behaviour modelling on academic performance, training aspirations and entrepreneurial success within career settings, with a more recently acknowledged impact upon women specifically within these settings [10], role models may also protect young people from the harmful physical or psychological consequences of stress by providing experiential information, advice or encouragement and therefore enable the development of emotion-focused coping mechanisms [11].

Besides role models, mentors can also play an important role in sexual and gender minorities’ lives. The term “mentor” has been evolving and changing over the last 3 decades [12]. However, in this scoping review, we use a general definition of mentorship. Generally, mentoring relationships occur when a less experienced person is being provided with advice, guidance and assistance by a more experienced and senior individual [12].

Although the role model and mentor (RMM) constructs have overlapping properties, the most significant difference is that role modelling does not require a direct relationship between people; the figure can be historical or famous [13]. Nevertheless, similar to role models, mentors can be associated with positive or negative psychological and physical outcomes as well as with personal and professional development [14]. Mentoring has been shown to be able to produce positive health behaviour changes and facilitate social networks whilst providing emotional security. These findings have been replicated across young people as well as adult populations [15]. However, in order to achieve positive change, mentoring should be adjusted to meet the needs of the individual. This is important, as minority groups, including SGMs, face unique stressors due to discrimination. Therefore, having a mentoring relationship tailored to the individual is important to reduce the disadvantages faced by minority populations [16].

Internal variation regarding the importance of role modelling relationships may also exist depending on the concordance of certain shared characteristics, such as race, ethnicity or gender between the mentee and mentor [17,18]; however, the significance of this variation for role models specifically is unclear. In a cross-sectional survey on mentoring in academic medicine conducted on junior doctors and faculty members, female doctors were significantly more likely to report that a same-sex mentor would be more understanding compared to their male counterparts. Interestingly, this finding was contradicted by faculty members, the majority of whom reported that it was not important to have a mentor of the same sex [19].

Despite emerging evidence regarding the importance of RMMs for some populations, there has been no previous review on the impact they have on the health and wellbeing of LGBTQ+ people. With an increasing awareness of the prevalence of SGMs in the UK [20] population, the U.S. LGBT population being estimated at 11 million Americans [21], and an ongoing mental health burden emphasising the vulnerability of this group, a greater understanding of the impacts that RMMs may have on this group is of great importance. This may help inform strategies for the prevention of mental or physical health difficulties in healthcare and social settings for LGBTQ+ people. This systematic scoping review aims to identify the experiences sexual and gender minority people have had with RMMs and explore the impact of this on their mental and physical health and wellbeing.

## 2. Materials and Methods

### 2.1. Protocol

This review has been reported according to the PRISMA (Preferred Reporting Items for Systematic Reviews and Meta-Analyses) 2009 checklist [22].

### 2.2. Eligibility Criteria

Participants: Any studies including sexual and gender minority people (e.g., lesbian, gay, bisexual, transgender, genderqueer, non-binary, pansexual, queer or questioning) were included. We defined sexual minorities as anyone who experiences sexual attraction to or has sexual contact with the same or more than one gender, and gender minorities, including transgender and non-binary people, as individuals whose gender identity or expression is different to their sex assigned at birth. We included studies that combined sexual and gender minorities and heterosexual, cisgender people, in which some results were available for sexual and gender minority participants.

Interventions and comparators: The population must have been exposed to role models, mentors, idols or heroes before or at the same time as the papers’ publication. When possible, studies with comparative data on those either not having a role model (or with a lack of exposure to role models) or from non-LGBTQ+ populations were included, though we did not exclude studies without comparator groups.

Outcomes: Any descriptions of the impact of role models on mentees’ health or wellbeing.

Study design: Any empirical paper (published and grey literature) available from the year 2000 onwards was included and our search was not limited to studies from any specific geographical location. The search was not limited by language.

Excluded were studies in which the outcomes for SGM sample and heterosexual sample were conflated, in which the mentoring or role modelling relationship did not reflect this systematic scoping review’s definition and in which the impact of the role modelling relationship was not measured or reported.

### 2.3. Information Sources, Search Strategy and Study Selection

The searches were conducted in May 2022. Databases (and platforms) searched were CENTRAL (Cochrane Library), ERIC (EBSCO), Science Citation Index (Web of Science), Scopus, Medline (Ovid), EPub (Ovid), Embase and PsychInfo. The same search strategy was used for each database but tailored when necessary.

Search terms were developed using Medical Subject Headings (MESHs) and key words, adding supplemental words to cover all areas of the systematic scoping review’s interest. These included synonyms for gender and sexual minority people, including the subgroups (such as lesbian and gay) combined with role model, mentor, icon or hero. All titles and abstracts identified via database searches were exported into EndNote Web, duplicates were identified and removed and remaining papers were screened by title and abstract by two researchers (JC and AD), who were blinded to one another’s decisions. Inclusion criteria were applied to each study, with any discrepancies being resolved through discussion between researchers. All papers identified through abstract and title screening were cross-checked by a senior researcher (KR).

### 2.4. Data Collection Process and Data Items

Data were collected by two researchers (JC and AD) working independently, and no automation tools were used. Discrepancies were resolved through discussion, and unclear information was addressed by a senior researcher (KR). To achieve systematic data extraction, two tables were created with the first table used to extract key study, population and experimental characteristics. The second table listed the outcome variables and any additional information on the analysis. Data were collected for comparator individuals or groups if applicable and for target groups.

Any type of outcome relating to health and wellbeing following exposure to RMMs was eligible for inclusion. These outcomes were grouped into broad domains following data extraction, based on common characteristics. For example, depression and anxiety scores post-exposure to role models were grouped into psychological wellbeing because both outcomes relate to mental health. The following domains were identified regarding the impacts of RMMs on LGBTQ+ people: psychological wellbeing and other mental health characteristics; substance use and misuse; sexual health; and coming out and identity acceptance.

### 2.5. Risk of Bias in Individual Studies

Critical Appraisal Skills Programme (CASP) checklists were used to assess risk of bias in included studies, and the relevant checklist that was appropriate for the study design of each study was chosen [23]. The tool has ten to twelve questions, including those pertaining to the appropriateness of methodological choice and rigorousness of data collection and analysis, to assess the overall risk of bias across three main domains. Two researchers (JC, AD) independently assessed the quality of included studies and discussed papers when conflicts arose. Conflicts not resolved through discussion were addressed by the research lead (KR).

### 2.6. Summary Measures and Data Analysis of Qualitative Data

Quantitative data were synthesised through narrative description and tabulation. We did not conduct meta-analysis due to heterogeneity in measures, limited data and our study design. For qualitative studies, two researchers (JC, AD) independently extracted and tabulated quotations from all relevant studies. Data were grouped into four separate domains suggested by one researcher (AD), which were then reviewed, revised and agreed upon by all researchers during the data extraction process for the identification of common themes. One researcher (JC) analysed themes from raw data and tabulated the comparative health impacts of positive, negative or absent role model figures. These were then independently appraised by all researchers, and inferences were added or expanded. The results were then analysed illustratively alongside selected quotations from the included studies. Inferences and conclusions following quantitative analysis and qualitative thematic analysis were combined in the discussion to provide an overview of the collated evidence.

## 3. Results

### 3.1. Study Selection and Characteristics

We found 501 citations, of which 78 were duplicates (see Figure 1 for a detailed PRISMA flow diagram). Subsequently, the title, abstract and full-text screening resulted in a further 412 exclusions. The most common reason for exclusion was outcomes not relevant for our review, such as the impact of role modelling or mentorship on educational attainment or employment status. There were 12 studies included, nine of which presented qualitative data [24,25,26,27,28,29,30,31,32] and three of which presented quantitative data [33,34,35].

The majority of the participants (n = 1468) were male and often from local youth groups or school settings. The age of included participants—mentees and people commenting on their RMMs—ranged from 15 [35] to 63 years [24]. There was a large variation in sample size, with some studies reporting a recruitment pool of six participants [28], and others as high as four hundred forty-seven (33). The impact of role modelling for men attracted to or displaying sexual behaviour with other men was reported in four studies [26,27,32,35]; in three studies, the participants were women attracted to other women [24,28,31]; three studies collected data on mixed SGMs [25,29,33]; and one study reported mixed sexual minorities (gay, lesbian or bisexual) [34]. None of the included papers reported the recruitment of intersex or asexual participants. A summary of the studies’ characteristics is displayed in Table 1.

### 3.2. Risk of Bias in Studies

The findings of the risk of bias assessment for the qualitative studies are summarised in Table 2, and for the quantitative studies, they are summarised in Table 3. With regards to the proportion of low- and high-quality studies in relation to how they contribute to the results, there is no association between study quality and quantity of contribution. The paper by Taylor and Snowden yielded the lowest methodological quality [31]. The papers highest in quality were those of Bird et al. (2011), Drevon et al. (2015) and Lauby et al. (2017). Regarding generalisability, Drevon et al. (2015) provided evidence for the need for natural mentorship to support positive educational outcomes for LGB youth. The authors explored the broader implications of mentoring, showcasing its potential to bring about positive changes at both the societal and individual levels.

### 3.3. Qualitative Study Results

A summary of the findings from each qualitative study is presented in Table A1 (Appendix A), with the identified themes of both the perceived positive and negative effects of RMMs, or their absence, summarised in Table 4. There were sixteen instances of participants referencing the impact of direct RMMs (participants with personal or one-to-one relationships with their role model), two accounts of the impact of indirect RMMs (icons, celebrities or television characters) and four accounts of the impact of not having an available or visible role model. The primary outcomes emerging from the qualitative data included the impact of RMMs on coming out and identity acceptance, psychological wellbeing, substance misuse and sexual health knowledge or practices.

#### 3.3.1. Coming out and Identity Acceptance

Seven studies reported outcomes relating to identity acceptance or coming out experiences [24,25,27,28,29,30,31]. The most common theme relating to this outcome was the importance of RMMs in terms of destigmatising LGBTQ+ identities or topics. Other themes included the impact of RMMs on participants’ ability to live authentically and openly with regards to their sexual or gender identity and RMMs affirming one’s sense of self.

*Destigmatising LGBTQ+ identities*. Four studies reported the impact of RMMs in destigmatizing LGBTQ+ identities [27,28,29,30], which was experienced with both direct and indirect role models. In French et al. (2009), one participant reflected on her exposure to direct role models:

“I think it just made me feel okay that there were gay and lesbian teachers and they were fine and they were the same basically as everyone else”. [28]

In this case, the presence of a role model with a visibly queer identity provided comfort to a young LGBTQ+ person in their school environment. In addition to normalising gay and lesbian identities, the public response of one role model to an incident of homophobic graffiti on university property provided comfort to one gay high schooler:

“The president of the university came forward—and… said ‘If the person… responsible for this decides to come forward, I will prosecute them myself… this university does not foster that kind of behavior… I support your community… this will not be tolerated.’… I’m glad that… this is being made into a big scene”. [29]

This was in contrast to their high school experience, where the participant comments on the absence of positive environments or role models in this educational setting, with the resulting effect being stigmatisation:

“It was something that they just swept under the rug, and you didn’t talk about it. Um, I never heard any anti-gay remarks from the teachers. It was always just the class [who made anti-gay comments], and that was it. But my school never had any kind of a gay-straight alliance or a safe space. It was just never talked about. And that’s the way it was”.

Indirect role models, such as celebrities and television characters, also played a role in destigmatising LGBTQ+ identities. Most commonly, this was by way of comfort through exposure to role models with shared identities. One participant in Forenza et al. (2017) notes the following:

“One piece of media that really hit home for me—and this is when I started thinking about (coming out)—was when (TV) started introducing gay characters or characters coming out of the closet”. [27]

In this instance, the participant cited an on-screen character as a positive role model for coming out as gay. In cases in which participants lacked these on-screen role models, they also expressed feelings of exclusion from cultural norms. Although there were no direct quotes available, the authors in French et al. (2009), a study looking specifically at the experience of lesbian women in science, technology, engineering and maths fields, report one participant as “[feeling] gay males were accepted in her school due to contemporary television shows like Will & Grace. She only recalls one openly gay female and felt at the time lesbians would not be as accepted as gay men. She feels her community at school did not have a place for lesbians” [28].

*Living Authentically and Openly as LGBTQ+.* A significant additional theme relating to coming out and identity acceptance was the impact role models have on the psychological safety of LGBTQ+ people to live authentically to their sexual or gender identity. Three studies reported this theme [25,29,31], with Briganze et al. (2001) only referencing role models as being helpful in the coming out process with no direct extractable data [24].

Multiple studies identified mentors as influencing their decision to remain closeted. For example, in Sadowski et al. (2009), one participant (herself a youth support group attendee) directly attributes not coming out to the fact her guidance counsellor was also not being openly gay, with this ultimately contributing to her displaying heteronormative behaviours:

“But she wasn’t out. So it was kind of hard to understand, like—and that was probably one of the reasons that kept me in the closet throughout high school… I also dated a couple boys… re-emphasizing the fact that I was straight… But I think one of the reasons I didn’t come out was the fact that she, my guidance counselor, didn’t either”. [29]

Donahue et al. (2007) also identified participants voicing the idea that their role model not being ‘visibly queer’ may have impacted their own coming out process:

“I guess I’m kind of the same way about it as she is. I kind of feel like I probably would have been anyway, but maybe not. Maybe if she had been visibly queer, maybe I’d be out, too”. [25]

For those without any role model (either positive or negative), the authors of the same study briefly note that two other participants ‘expressed uncertainty about whether having an ‘out’ teacher role model would have led to their being more open in the school’. In other studies, although there is no direct reference to the impact of having an absent role model, some participants attribute the absence of advice as a key factor in the suppression of their sexuality [31]:

“To recognize that maybe I felt a little bit out of place and I could’ve used some extra counsel…I think just encouraging me to be a little more open [about my sexuality] in college probably would have changed my experience a lot… I just wanted to suppress my sexuality in college”.

*Affirming Sense of Self*. Two studies reported the impact of affirming role models or mentors [27,31]. Sources of affirmation were commonly family members or positive messages through media or celebrities. One participant references the importance of affirming messages from their pop icon:

“Right around when I was coming out myself, Lady Gaga came out with her Born this Way album. It just seemed like perfect timing—that it was the time I decided to come out—and it was such a revolution for [Lady Gaga] to say that you’re beautiful and brave to be the way you were born”. [27]

Another participant of the same study reflects that their role model provided affirmation, but they did not have the language or understanding to describe this at the time:

“I didn’t know what gay was—I didn’t know what gay affirming was… but I knew that they were”.

#### 3.3.2. Psychological Wellbeing

Four studies reported outcomes relating to the impact of role models or mentors on psychological wellbeing [27,29,30,31]. The themes emerging within these outcomes included positive role models reducing stress, as well as increasing self-confidence, mood and psychological safety, with negative role models leading to feelings of reduced psychological safety and/or stigmatisation.

*Stress Reduction*. Three studies presented data that pertain to the mechanisms by which mentors affect stress, with the mentoring relationships occurring in two educational settings (medical school and high school). Within medical school, where supervision and mentorship are mandatory for professional development and progression, some participants felt that having a shared identity with their mentor would have been beneficial:

“I think someone from the community would be a little bit more desirable because being LGBT would be one less thing to explain and have to deal with or stress over”. [30]

Role models with concordant identities also positively impacted participants’ mood, as one participant puts it:

“Even though I wasn’t [out as] gay,” he began, “it just gave me so much joy to see someone so campy and over-the-top that I could kind-of relate to”. [27]

In addition to the identity concordance of mentors in itself having a role in stress reduction, mentors also played a role in stress reduction via the provision of empathy, advice or support:

“She’s a great person. She listens… she gives great advice… because she actually knows the things you go through… you know she cares… I’ve always seen her like a guardian… she watches over me”. [31]

Personal proximity to issues experienced by SGMs was also cited as important, as mentors could provide additional perspectives in circumstances in which discrimination is unclear [30]:

“I think it’s important for me to have at least one LGBT mentor because… being a sexual minority and a racial minority I tend to have a lot of internalized stigmas. Like, whatever negative is happening in the situation, is it because of my sexual orientation, my gender identity, or my race?… It’s important to have someone who can sit down and say no, it’s not that, it’s actually this. Or to share in that experience and to say this is how I overcame it and this is how you can overcome it and keep moving forward. That is something that’s invaluable”.

*Confidence Increasing or Decreasing*. One study reported participants’ experiences of mentorship and its impact on their confidence [31]. Positive experiences with mentors are found to improve participants’ confidence by way of affirmation:

“She helped me be more confident into who I am… She told me like it does not matter who I like-that my sexuality matters. That was good for me”.

*Psychological Safety Increasing or Decreasing*. Psychological safety is feeling that it is safe to express oneself intellectually, emotionally and socially and describes the ease at which an individual can take interpersonal risks (Newman et al., 2017). Three studies reported outcomes relating to participants’ perceived psychological safety within their respective environments whilst in the presence of their mentor [29,30,31]. Psychological safety was commonly referred to by participants in spiritual or religious environments. In describing her mentoring relationship at church, one participant in Taylor and Snowden (2014) notes the following:

“I am sort of mentored by an older woman at church who is married and had a family… and I found I was able to sort of discuss my feelings on sexuality and sort of where I felt I sat and my perspective on what the church was doing… so that, I think, was very valuable to me that…there was someone that I could discuss that with, someone who was a Christian and in the church who got that and so I found that very helpful”. [31]

This participant proposes that her mentoring relationship was valuable to her because it allowed her to express her thoughts without fearing consequences. In contrast, direct experiences of homophobia from mentors may result in some LGBTQ+ people feeling uncomfortable expressing themselves. One dental student states as follows:

“You go to a mentor, you want to be comfortable with them. If they are homophobic in any way then you would obviously not be comfortable in divulging any kind of personal information to them. You just wouldn’t be comfortable in their presence, and they would not be able to help you in any way. If they aren’t comfortable with who you are, your core, they wouldn’t be able to mentor you in any way”. [30]

#### 3.3.3. Substance Use and Abuse

Only one study reported qualitative outcomes relating to substance use and abuse (31). The most common impact cited by participants was that mentors impart knowledge based on their experiences to try to help them not take part in risky substance use behaviours. In Taylor and Snowden (2014), one participant explains as follows:

“I did have a mentor growing up… [he] protected me um, from a lot of the like “vices”… I had someone take me under their wing and like kind of teach me that there were certain things that I needed to know, to be aware of, and not to go down the wrong path. ” (Kyle, European American, gay, 22 yrs. old) [31]

#### 3.3.4. Sexual Health

Two studies presented qualitative data pertaining to the impact of role models or mentors on sexual health [26,31]. Both studies reported that role models or mentors increase sexual health knowledge.

Participants reported that their sexual practice was positively impacted by RMMs, namely by increasing awareness of sexually transmitted infections, as well as preventative measures for STIs and encouragement of sexual health testing. One participant who was part of a peer mentor modelling programme describes the impact their peer had on them:

“We are the abandoned groups of people in this community…no one really care about our life and existence. But my peer also encouraged me to start my own direction… I was lost… I don’t know how can I protect myself away from unsafe sexual activities… no one cared me at all…due to this Peer [Mentor] Modelling Programme… I know what to do now”. [26]

This participant believed that their mentor benefitted them through both increased guidance as well as through the provision of emotional support and care. The sentiment of mentors being helpful in this context due to their ability to provide guidance was also mirrored by another participant:

“I had a lifelong friend… older than me… gave me, like advice, go protect yourself, HIV awareness, go get checked out… it’s like they beat it into me so much that now I do get checked up on a regular [basis]”. (John, Latino, gay, 21 yrs. old) [31]

### 3.4. Results of Quantitative Studies

A summary of the findings from the three studies reporting quantitative data (33–35) can be found in Table 5. Outcomes included the impact of role models, mentors or mentoring programmes on binge drinking, drug use, psychological distress, suicidal ideation and the risk of contracting sexually transmitted infections.

#### 3.4.1. Alcohol and Substance Misuse

Two studies reported the impact of role models or mentorship on binge drinking or problems caused by alcohol and substance misuse (n = 894) [33,34]. In Drevon et al. (2015), there was no significant impact of mentorship on LGB individuals with illegal drug use or problems caused by alcohol. There was, however, a significant impact on high school exit status (*p* < 0.01, with OR 2.51, 95% CI 1.08–5.80). In Bird et al. (2011), there were no significant differences between those with role models and those without role models with regards to binge drinking or drug use.

#### 3.4.2. Psychological Distress

Two studies reported the impact of role models or mentorship on psychological distress (n = 894): one on psychological distress, broadly speaking, as the presence of anxiety or depressive symptoms [33] and another as the prevalence of suicidal ideation [34]. In Drevon et al. (2015), there was no significant impact of mentorship on suicidal ideation. In Bird et al. (2011), there was a significant difference in psychological distress in those without role models and those with role models, contrary to the findings that they expected. Following subgroup analysis on role model accessibility using post hoc Bonferroni tests, there was an increased risk of psychological distress in LGBT individuals with inaccessible role models compared to that of those with accessible role models or no role models at all.

#### 3.4.3. Sexual Health Risk

Two studies reported the impact of role models or mentorship on sexual health risk (n = 720) [33,35]. In Bird et al. (2011), there was no significant difference in STI diagnosis between those with role models and those without role models. This was also true when role model accessibility was assessed [33]. In Lauby et al. (2017), the risk of developing HIV in terms of participants’ ‘anal sex risk score’ prior to and following a 36-month community mentoring programme was calculated in two separate city populations: Philadelphia and Baltimore [35]. In the Philadelphia programme, there was a significant decrease in HIV risk (*p* < 0.02) post-programme compared with that pre-programme, whereas in Baltimore, the anal sex risk score increased following the intervention (*p* < 0.02).

## 4. Discussion

### 4.1. Summary of Results

This systematic scoping review is the first to collate data assessing the association of RMMs or mentoring programmes with the health and wellbeing of SGMs. The included studies were mostly cross-sectional in design, qualitative and limited in sample size and methodological quality. The results show that positive RMMs are correlated with a number of psychosocial and physical outcomes affecting mixed SGMs, including the coming out process, identity self-acceptance, psychological wellbeing, substance use and sexual health. This is true for the case of positive role modelling or mentoring relationships, regardless of setting and proximity, with a corresponding negative impact in cases in which role models display negative behaviour or in which role models are absent or inaccessible. Proposed mechanisms for how positive RMMs may influence SGM health and wellbeing are displayed in Figure 2.

We found that RMMs are commonly perceived to influence coming out experiences, sexual or gender identity self-acceptance and feelings of acceptance by others. This occurs mainly in relation to stigmatisation, but also in the affirmation of LGBTQ+ identities. RMMs enable destigmatisation by setting boundaries regarding the tolerance of LGBTQ+ phobia. Unfortunately, some potential role models increased stigmatisation by not challenging homophobic behaviours or by avoiding discussions of LGBTQ+ issues. Ways in which mentors display behaviours that induce or suppress stigma may provide an indirect explanation for multifactorial variables which contribute to minority stress, with stigma itself remaining a high predictor for poor health outcomes amongst LGBTQ+ people [6]. In cases in which RMMs are absent, there may be fewer mechanisms by which this destigmatisation can occur.

We found that indirect role modelling through celebrity or icon exposure through media provides similar access to positive affirmative messaging, as in the case of direct role models. Existing research on the role of positive affirmative media messaging on the mental health of LGBTQ+ people reflects this finding; frequent exposure to negative depictions of transgender adults within the media is associated with significantly higher risks of depression and global psychological distress [36]. This is also true amongst sexual minority populations in societies in which there is little LGBTQ+ media representation [37].

Participants noted the benefits of role models who were able to live openly and authentically. However, existing data regarding the psychological impacts of SGM people remaining closeted from others are conflicting. Recent studies investigating the mental health of sexual minority people before and after coming out identified that out sexual minority men were significantly more likely to have major depression than closeted men, though this pattern is reversed in women [38]. It has been suggested that the concealment of one’s sexuality may enable sexual minorities to avoid discrimination and its associated impact on mental health [39], with the impact being more significant than the guilt and shame associated with the concealment of one’s sexuality from others.

Mechanisms by which psychological wellbeing may be influenced by RMMs are highlighted. We found that shared role models may affect psychological wellbeing by reducing stress, increasing or decreasing confidence and providing or failing to provide psychologically safe environments for SGM people. Having a shared identity with role models or mentors is a key factor for SGM people in reducing stress. The shared understanding of experiences by a mentor or role model reduces barriers in having to justify experiences of discrimination and provides insight into whether mentees’ life experiences are related to their minority status. Existing research into the impact of mentor–mentee concordance within other minority groups at high risk of prejudice and discrimination (such as racial and gender minorities) supports this, but these data are largely restricted to academic and medical settings. Regarding gender matching, particularly in under-represented medical fields, concordance increases mentoring satisfaction [40,41]. In cases of racial minorities, barriers to effective mentorship, as similarly described in this review, may stem from ‘having to explain their culture to their mentor’ with whom they do not identify [42], contributing to existing stress. Further research into identity-specific challenges to mentorship amongst LGBTQ+ populations may further inform this theory.

An unexpected theme regarding mentorship and wellbeing highlighted RMMs’ roles in psychological safety. Psychological safety is one’s level of comfort in expressing themselves intellectually, emotionally and socially, therefore influencing interpersonal risk-taking behaviours [43]. This internal dynamic is thought to particularly affect minority groups, who are less represented in leadership and management positions. For organisations, improving psychological safety is also the factor most associated with increased productivity [44]; but for individuals, a high level of psychological safety may reduce internal team conflict and improve wellbeing. For LGBTQ+ mentees, psychological safety is largely under-researched, but here we identify that RMMs play a role in mentees being able to discuss their sexuality without consequence, which may indirectly improve their mental wellbeing. This warrants further research.

There was no significant evidence of the impact of RMMs on substance use. Although one study [31] reported a participant suggesting that mentors provide a protective effect against substance use, this relationship was not supported by the available quantitative data. This is in contrast to existing research amongst heterogeneous populations of LGBTQ+ and non-LGBTQ+ people, in which natural mentoring relationships are thought to improve perceptions of a mentee’s perceived coping ability and therefore indirectly reduce cigarette, alcohol and marijuana use [45]. Additional research on the effect of natural mentoring relationships on SGMs’ substance use is needed to corroborate these findings, given the unique challenges that these minorities face.

Lastly, qualitative data identified RMMs and mentoring programmes as vectors in increasing sexual health knowledge, predominantly by passing on RMMs’ own experiential knowledge to improve mentees’ sexually transmitted infection awareness, protection advice and sexual health screening. There was not sufficient quantitative evidence to substantiate this further, but the acquired qualitative data match existing research regarding the role of role models in the encouragement of knowledge acquisition, awareness and engagement in sexual health [46].

### 4.2. Strengths and Limitations of Review

The inclusion of diverse studies including academic journals, reports and grey literature broadens the findings’ scope and generalizability. The scoping review includes a CASP quality assessment of the included studies in order to evaluate the validity of the employed research methodologies, augmenting the credibility of the research [23]. We incorporated an extensive search strategy reviewing both qualitative, quantitative and mixed design studies. These results were thoroughly analysed and synthesised. Additionally, a model was also developed that outlines the mechanisms by which role modelling and mentoring can influence health outcomes.

To assess the quality of cross-sectional data, we used the CASP checklist for cohort studies [23]. While certain assessment points of the cohort study checklist regarding follow-up, such as “Was the follow up of subjects complete enough?”, were not applicable to cross-sectional data, we found that the tool was effective in identifying recruitment biases that could limit generalisability. Future reviews could include educational impacts and how psychological well-being and educational outcomes may be related for these groups.

As the definition of role model may encompass different terms such as parents or religious figures, this review is limited in its ability to operationalise RMMs for search purposes. The lack of consensus of the definition of RMMs is acknowledged by existing research [13].

### 4.3. Limitations of Included Studies

The sample sizes of some reviewed papers were not enough to adequately support the research question. Many studies used the non-probability convenience sampling method and recruited participants from LGBTQ-specific institutions and events. For example, in the study by Torres et al., data were collected from community-based agencies and venues used by same-sex attracted youth in Chicago, which may be not representative of the whole same-sex-attracted youth population [32]. Youth attending these venues and events are more likely to be out and to be comfortable being associated with LGBTQ identities. These groups may also show different role modelling outcomes to youth that do not attend SGM events.

The studies did not account for intersectionality or how various marginalised identities may influence role modelling outcomes. Characteristics such as gender, sex, sexual identity, race, age, socioeconomic class and disability form complex and interconnected systems and can result in unique extra stressors for an individual [47]. Therefore, due to stressors stemming from different discrimination and victimisation processes, people with intersectional marginalised identities may need more support from RMMs and/or could benefit more if this is received. In addition to this, there is an unequal representation of different LGBTQ+ subgroups within the SGM literature, with men who have sex with men dominating the field. The included papers did not collect data on asexual and pansexual people, so they were not able to report role modelling outcomes specific to subgroups due to the lack of evidence on less-researched populations of the SGM community.

As a result of the above, in combination with the lack of control groups within each study, this scoping review cannot infer cause and effect for any potential influence of RMMs on SGMs.

### 4.4. Implications for Policy

The findings of this scoping review have significant policy implications, highlighting the potential of role modelling strategies in fostering acceptance and promoting the health of the LGBTQ+ community. Mentoring programmes implemented in various settings are effective strategies for reducing stress among SGM groups. They may also increase psychological safety in the workplace or in education. This scoping review revealed that RMMs play a crucial role in improving SGMs’ psychological safety, especially in spiritual or religious environments in which individuals with diverse sexual and gender identities face increased stigma. Mentoring-centred policies can promote diversity and inclusion within organisations, ensuring that SGMs are adequately represented. By doing so, these policies not only increase the visibility of underrepresented SGM groups but also help de-stigmatize their identities, thereby encouraging individuals to live openly and authentically as LGBTQ+.

Policies around mentoring can provide a significant knowledge base to the SGM community. RMMs use their personal experiences to pass on valuable knowledge, particularly in the context of discouraging risky substance use behaviours among the SGM population. In addition, the incorporation of SGM mentors has been shown to increase sexual health knowledge, potentially leading to a decrease in the prevalence of STIs in the community.

### 4.5. Implications for Further Research

Given the lack of representation of some groups within the SGM community and intersections, prospective research should incorporate rigorous data collection methods that encompass all SGM subgroups as well as intersectional identities.

It would be valuable for research to investigate how role model characteristics impact health outcomes, in the same way that a study on 1013 students examined whether gender and race matching have an impact on academic performance and what students think of the importance of gender and race congruence [48]. Although participants reported that matching is important to them, and those with gender-congruent mentors reported greater perceived levels of help, academic performance was not significantly correlated with gender and race matching. However, another study found that gender matching was associated with a significant improvement in academic achievement and performance in a sample of students [49]. Thus, it is essential to conduct future cohort studies with a large sample size to test whether gender, sexuality, race matching and age differences have an impact on role modelling outcomes.

Future studies could investigate further how RMMs improve psychological well-being and sexual health knowledge among SGMs, which can inform the development of interventions aimed at reducing health inequalities among this population.

Additionally, this review reports a simplified view of the perceived impact of RMMs on health and wellbeing and does not explore whether apparent perceptions of ‘no role model impact’ may be a result of conflicting positive and negative forces. Future interview-based research could explore RMMs as multifaceted positive and negative forces influencing different aspects of health and wellbeing.

## 5. Conclusions

Within the confounds of the limitations of this study and the included papers, our findings suggest that role models may have a role in improving psychological wellbeing by destigmatising LGBTQ+ identities, reducing stress, improving confidence and providing psychologically safe environments for sexual and gender minority people to exist. It also highlights possible mechanisms by which discriminatory behaviours or negative attitudes displayed by role models and those in mentoring positions lead to stigmatisation, the concealment of one’s sexuality and internalised shame. In this regard, our findings also support existing research on the health impacts of minority stress [4,5,6] on SGMs and provide an explanation for additional vectors by which this marginalisation occurs (i.e., through negative role models). Thus, this study may provide an incentive for matching initiatives within educational settings for SGM people and role models with similar experiences of marginalisation or empathy towards marginalised people, as well as schemes to increase the visibility of open LGBTQ+ role models or mentors. This mirrors existing research in relation to gender- and race-matching initiatives for other marginalised groups [40,42]. Further research into matched mentoring programmes and specifically the mechanisms by which role models influence health is imperative to assist in the implementation of such policies.

## Figures and Tables

**Figure 1 behavsci-14-00417-f001:**
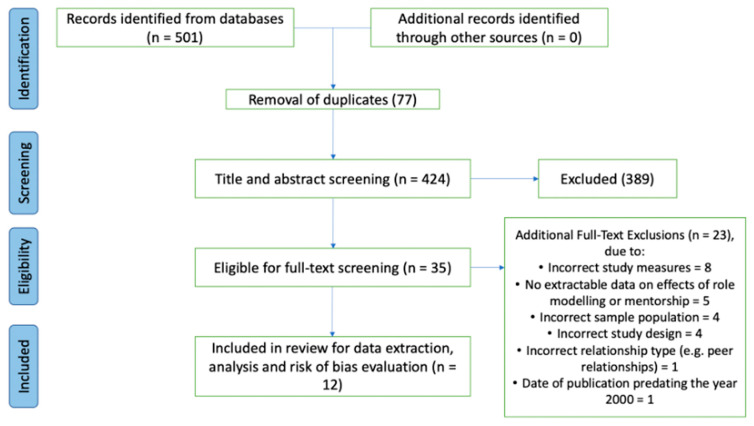
PRISMA (Preferred Reporting Items for Systematic Reviews and Meta Analyses) screening flow chart.

**Figure 2 behavsci-14-00417-f002:**
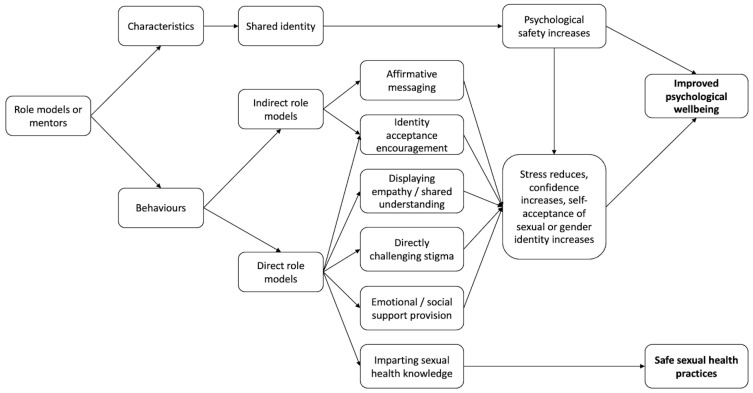
Proposed mechanisms by which role models or mentors impact mental and physical wellbeing.

**Table 1 behavsci-14-00417-t001:** Characteristics of included studies.

Study Author, Year	Study Design	Study Aims	Setting of Role Modelling	Recruitment Strategy	Participant Number and Characteristics	Role Model Characteristics	Outcomes of Interest
Bird et al., 2011 [33]	Cross-sectional survey	Investigate impact of accessible role models on LGBT health outcomes	Various settings dependent on RM accessibility	Convenience sampling from community LGBT youth agencies, high school postings, palm cards in gay-identified neighbourhoods, participant referrals	A total of 447 LGBTQ+ youth aged 16–24	Role models either accessible (parents, friends) or inaccessible (celebrities, historical figures)	Psychological distress, illicit drug use, self-reported STI diagnosis, anxiety, depression
Bringaze et al., 2001 [24]	Cross-sectional survey	Identify factors that lesbian leaders and role models found helpful in coming out process	Not reported	Participants selected if they were affiliated with national LGB organisations and/or were listed in “The Gay and Lesbian Address Book (1995)”	A total of 62 lesbian leaders or role models aged 24–63	Role models NOS	Helpfulness of role models and other factors in coming out process
Donahue, 2007 [25]	Open-ended interview and cross-sectional survey	Identify role that lesbian and gay teachers played in identity negotiation of student teachers	School	Not specified; likely convenience sample recruited from Mills College	A total of 13 gay and lesbian student teachers	Senior gay and lesbian teachers as role models	Importance of seeing a role model in terms of identity acceptance, coping mechanisms and openness in discussing identity with others
Drevon et al., 2015 [34]	Cross-sectional survey and in-home interviews	Examine instances in which natural mentoring relationships are associated with educational, career, psychological or behavioural (substance use) outcomes in LGB people	School and others (setting of role modelling NOS)	Used datasets including transcripts from National Longitudinal Study of Adolescent Health, and Adolescent Health and Academic Achievement (AHAA)	A total of 447 people (main cohort) and 409 people (high school exit sample)	Natural mentors (non-parental adults)	Education and employment, psychological well-being (self-esteem, depression, suicidal ideation), substance use and abuse
Dos Santos, 2021 [26]	Semi-structured interview, focus group	Promote health of gay male youths through a mentor modelling programme, and understand how mentoring relationships influence sexual health knowledge	Social media and fortnightly face-to-face meetings	Not reported	A total of 8 health and social care professionals and 40 gay male youths	Role model, mentor	Sexual health knowledge
Forenza, 2017 [27]	Semi-structured interview	Examine importance of gay icons and the coming out process for out gay men	As represented in the media	Recruitment flyers circulated on social media (Twitter and other gay-affirming sites)	A total of 10 gay men in their 20 s to 50 s	Icons (public entertainers with support for young gay fans)	Impact of icons on feelings of shared identity, sense of self and confidence in coming out process
French, 2009 [28]	Semi-structured interview and cross-sectional survey	Explore how the interest of marginalised groups (lesbian women) can be increased in STEM	Various (high school, college, medical school, home environment)	Purposeful sampling of 2 lesbian university-level students, 2 who had recently completed science majors and 2 with established careers in science	A total of 6 lesbian women in or beyond higher education	Role model or mentors (various: parental relationships, academic mentors)	Effects of RMs on career aspirations or decision making, interest in science, destigmatising homosexuality
Lauby et al., 2017 [35]	Semi-structured interview and cross-sectional survey	Evaluate effectiveness of mentoring programme on alcohol or drug use and HIV risk behaviours among young MSMs	Philadelphia and Baltimore	Longitudinal sample from those responding to advertisement, followed by respondent-driven additional sampling	A total of 273 young gay MSMs aged 15–27	Role models NOS	Changes in risk of developing HIV
Sadowski et al., 2009 [29]	Cross-sectional survey with open-ended interview	Investigate impact of relational connections on LGBTQ+ lives	Various (home, youth group, school)	Recruited directly by researchers attending meetings of 2 different LGBTQ youth support groups and offered questionnaire and/or interview of 1 h in length	A total of 30 LGBT youth attending youth support group	Role models or mentors NOS	Destigmatisation of LGBTQ topics, social support (feeling listened to, perceived openness), coming out experience
Sanchez et al., 2018 [30]	Cross-sectional survey with focus group analysis	Explore obstacles to and facilitators of successful mentorship for LGBT health professional trainees	Academic institution or healthcare setting	Convenience sample; email sent to those attending LGBT conference	A total of 117 LGBT health professional trainees	Mentors; “persons responsible for career and/or personal development of mentee”	Benefits of mentors on career and personal development
Taylor and Snowden, 2014 [31]	Semi-structured interview, diary analysis	Examine young lesbian Christians’ experiences of gendered and heteronormative role models	Church	Invitation to interview sent via advertisement on website of researchers, closed Facebook group (Queer Religious Youth), churches, LGBT university societies and support services	A total of 16 lesbian Christian youth	Role model and mentors specifically including male or female bishops, priests, vicars and lesbian religious leaders	Destigmatisation of LGBTQ topics, feelings of acceptance or being able to discuss feelings on sexuality
Torres et al., 2012 [32]	Cross-sectional survey	Identify the presence, form and function of natural mentoring relationships amongst GBQ men	Not reported	Convenience sampling from community-based agencies in Chicago often visited by gay and bisexual men	A total of 39 gay, bisexual or questioning men	Natural mentor	Social and informational support, emotional support, acceptance

Abbreviations: LGBTQ+: lesbian, gay, bisexual, transgender, queer or questioning and other non-specified identities; MSM: men who have sex with men; STEM: science, technology, engineering and maths; GBQ: gay, bisexual or questioning; NOS: not otherwise specified.

**Table 2 behavsci-14-00417-t002:** CASP (Critical Appraisal Skills Programme) assessment for qualitative studies.

Study Author, Year	1	2	3	4	5	6	7	8	9	10
Bringaze et al., 2001 [24]	Y	Y	Y	Y	Y	CT	N	N	Y	CT
Donahue, 2007 [25]	Y	Y	Y	Y	Y	N	CT	N	CT	Y
Dos Santos, 2021 [26]	Y	Y	Y	CT	Y	N	Y	Y	Y	Y
Forenza, 2017 [27]	Y	Y	Y	Y	Y	Y	CT	Y	Y	Y
French, 2009 [28]	Y	Y	Y	Y	Y	Y	Y	Y	Y	Y
Sadowski et al., 2009 [29]	Y	Y	Y	CT	Y	N	CT	CT	Y	CT
Sanchez et al., 2018 [30]	Y	Y	Y	CT	Y	N	Y	Y	Y	Y
Taylor and Snowden, 2014 [31]	Y	Y	Y	CT	N	N	CT	N	N	N
Torres et al., 2012 [32]	Y	Y	Y	CT	Y	Y	Y	Y	Y	Y

The checklist questions included the following: 1. Was there a clear statement of the aims of the research? 2. Is the qualitative methodology appropriate? 3. Was the research design appropriate to address the aims of the research? 4. Was the recruitment strategy appropriate to the aims of the research? 5. Were the data collected in a way that addressed the research issue? 6. Has the relationship between researcher and participants been adequately considered? 7. Have ethical issues been taken into consideration? 8. Was the data analysis sufficiently rigorous? 9. Is there a clear statement on the study’s findings? 10. How valuable is the research? Y = Yes, N = No, CT = Cannot tell.

**Table 3 behavsci-14-00417-t003:** CASP assessment for quantitative (cohort) studies.

Study Author, Year	1	2	3	4	5a	5b	6a	6b	9	10	11
Bird et al., 2011 [33]	Y	Y	Y	Y	Y	Y	Y	Y	Y	Y	Y
Drevon et al., 2015 [34]	Y	Y	Y	Y	Y	Y	Y	Y	Y	Y	Y
Lauby et al., 2017 [35]	Y	Y	Y	Y	Y	Y	Y	Y	Y	Y	Y

The checklist questions included the following: 1. Did the study address a clearly focused issue? 2. Was the cohort recruited in an acceptable way? 3. Was the exposure accurately measured to minimise bias? 4. Was the outcome accurately measured to minimise bias? 5a. Have the authors identified all important confounding factors? 5b. Have they taken into account the confounding factors in their design and/or analysis? 6a. Was the follow-up of subjects complete enough? 6b. Was the follow-up of subjects long enough? 9. Do you believe the results? 10. Can the results be applied to the local population? 11. Do the results of this study fit with other available evidence? Y = Yes, N = No, CT = Cannot tell.

**Table 4 behavsci-14-00417-t004:** Summary of identified themes of the effects of role models or mentors from qualitative studies.

Perceived Impact of Absent Role Models or Mentors by Mentees	Perceived Negative Impact of Role Models or Mentors by Mentees	Perceived Positive Impact of Role Models or Mentors by Mentees
Destigmatisation of LGBTQ+ identities prevented due to lack of representation The practise of concealing one’s sexuality from others continues	Internalised shame within mentees, with closeted role models discouraging their coming out Stigmatisation by role models expressing LGBTQ+-phobic attitudes Acceptance of identity from others reduced Negative coming out process: role model not supporting their way of life due to incompatibility within social setting (church), resulting in lack of confidence in securing leadership positions Perceived psychological safety decreases	Destigmatisation of LGBTQ+ issues and identities Love and acceptance from others increase Creating affirming environments Self-confidence increases Psychological safety increases Stress reduction by encouraging one to live authentically or to not hide one’s identity Sexual health knowledge increases

**Table 5 behavsci-14-00417-t005:** Quantitative results.

Study	Group Definition	Outcomes Measured	Result Target Group 1 (Mean (SD)) (N)	Result Target Group 2 (Mean (SD)) (N)	Result (Comparator) (Mean (SD)) (N)	Statistical Tests Used	Comparative Statistics and *p* Values
Bird et al., 2011 [33]	Role model vs. no role model	Binge drinking	2.33 (0.08) (n = 267)	NA	2.22 (0.08) (n = 180)	ANCOVA score Pearson’s chi-square	*p* = NS
		Drug use	0.76 (0.08) (n = 267)	NA	0.63 (0.10) (n = 180)		*p* = NS
		Psychological distress	60.2 (0.65) (n = 267)	NA	56.4 (0.78) (n = 180)		*p* = NS
		STIs	0.49 (0.06) (n = 267)	NA	0.49 (0.07) (n = 180)		*p* = NS
	Accessible vs. inaccessible vs. no role model	Binge drinking	2.22 (0.12) (n = 89)	2.39 (0.09) (n = 160)	2.22 (0.08) (n = 180)		*p* = NS
		Drug use	0.63 (0.14) (n = 89)	0.82 (0.09) (n = 160)	0.63 (0.10) (n = 180)		*p* = NS
		Psychological distress	58.0 (1.13) ºº (n = 89)	61.2 (0.79) (n = 160)	56.4 (0.78) º (n = 180)		º *p* < 0.01 ºº *p* < 0.05
		STIs	0.46 (0.10) (n = 89)	0.50 (0.06) (n = 160)	0.49 (0.07) (n = 180)		*p* < 0.01
Drevon et al., 2015 [34]	Role model vs. no role model	Risk of dropping out of school	NR	NA	NR	Linear regression and logistic regression	OR 2.51 (CI 1.08–5.80), *p* < 0.05
		Risk of suicidal ideation	NR	NA	NR		OR 1.19 (0.31–2.32)
		Risk of illegal drug use	NR	NA	NR		OR 0.79 (0.49–1.90)
		Risk of problems caused by alcohol consumption	NR	NA	NR		OR 0.96 (0.49–1.90)
Lauby et al., 2017 [35]	Prior to community mentorship programme vs. 36 months following programme	Risk of developing HIV in Philadelphia	44.73 ** (n = 164)	NA	67.24 * (n = 164)	Repeated measures analysis of variance (ANOVA); sociodemographic and health factors controlled	*p* < 0.02
		Risk of developing HIV in Baltimore	43.15 ** (n = 80)	NA	37.92 * (n = 80)		*p* < 0.02

NR = not reported, NA = not applicable, NS = not statistically significant. Higher scores indicate higher levels of reported outcomes. * risk of HIV score pre-mentorship programme; ** risk of HIV score post-mentorship programme. º *p* < 0.01; ºº *p* < 0.05.

## Data Availability

Not applicable, as this is a scoping review paper.

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
