# Peer review of "The Impact of Role Models and Mentors on the Mental and Physical Wellbeing of Sexual and Gender Minorities"

_behavsci, 2024, doi:10.3390/bs14050417_

Round 1

Reviewer 1 Report

Comments and Suggestions for Authors

1.  This is a well written and clearly explained research report.

2.   The references are not consistent but should be, mainly on how capitalization is used in titles of journal articles.

3.  Some of the non-significant results might be the result of conflicting forces, positive and negative, that combine to an apparent no effect.

Author Response

I would like to thank reviewer 1 for your comments on our article. In response to your review points:

  1. N/A
  2. References have been adjusted for consistent capitalisation amongst journal article titles and journal names as per the Chicago referencing guide
  3. Thank you for your input. We have included a comment in the implications for further research which may address causes for these non-significant results, see line 629.

Reviewer 2 Report

Comments and Suggestions for Authors

The manuscript “The Impact of Role Models and Mentors on The Mental and Physical Wellbeing of Sexual and Gender Minorities: A Systematic Scoping Review” focuses on an interesting and important topic. It is very well written and clear, for this reason I think it should be accepted for publication.

I just make some suggestions for improvement. In fact, I suggest that you carefully read the manuscript one last time, because there are some errors, probably due to familiarity with the text. I highlight the following:

There is an error in the title of section 3.3. Qualitative Study Ressults (line 236).

In line 246, do you mean Figure 2 or Table 2?

Pay attention to the table numbers, which are not correct as they are: where you say Table 3 you should say Table 2; where you say Table 4 you should say Table 3; where you say Table 2 you should say Table 4; where you say Table 6 you should say Table 5.

Finally, I suggest that when you present Table A1 (on line 237), you orientate your readers and refer to Appendix A.

Author Response

Thank you reviewer 2 for your helpful comments. I have amended the table and figure numbers so they are correctly referenced in text and in chronological order. 

In response to the query regarding 'figure 2' on line 246: it is correctly referencing figure 2, however I appreciate this may have been unclear from the introductory prose. I have removed this sentence as the figure is more appropriately addressed in the discussion section, see line 475.

Reviewer 3 Report

Comments and Suggestions for Authors

Introduction:

1.        Please use a coma after “and” when using more than two adjectives. 

2.        Please check for the titles and subtitles upper and lowercases. 

Results:

1.        Table 2 needs to be revised. Columns are not clear. I suggest not adding bullets to the table. 

2.        Table 6. Why are ‘n’s empty? 

Discussion 

1.        Please include more references to strength the conclusion. 

Comments on the Quality of English Language

Looks good to me. My only concern was to include a coma after “and” when using more than two adjectives. 

Author Response

Thank you reviewer 3 for your helpful comments. I have addressed responses to these below:

Introduction

  1. An attempt has been made to edit the text to ensure the use of the comma is grammatically correct; if this is not sufficient, or detracts from the overall quality of the paper, please can you aid on specific examples of this error. Familiarity of the text may hinder correction here.
  2. Title and subtitle capitalization has been edited so it is consistent throughout the text.

Results

  1. Bullet points have been removed as suggested; please inform me if this has not been done as it is unclear when tracked changes are enabled.
  2. I have added perceived impact of X "by mentees" in line 250 to aid in clarity of interpreting the table.
  3. The 'N's in table 6 (now labelled Table 5 due to labelling error) have now been inputted, with thanks for noticing.

Conclusions 

  1. Additional references have been added to the conclusion as requested.